



# Assessment of methane emissions from US onshore oil and gas production using MethaneAIR measurements

Katlyn MacKay*[1,2], Joshua Benmergui[1,2,3], James P. Williams[1,2], Mark Omara[1,2], Anthony Himmelberger[2], Maryann Sargent[3], Jack D. Warren[1,2], Christopher C. Miller[1,2,3], Sébastien Roche[1,2,3], Zhan Zhang[3], Luis Guanter[1,2], Steven Wofsy[3], Ritesh Gautam*[1,2]

[1]Envionmental Defense Fund, New York, NY, USA 10010
[2]MethaneSAT, LLC, Austin, TX, USA 78701
[3]Harvard University, Cambridge, MA, USA 02138

*Correspondence to*: Katlyn MacKay (kmackay@edf.org) and Ritesh Gautam (rgautam@edf.org)

**Abstract.** Mitigation of methane emissions from the oil and gas sector is an effective way to reduce the near-term climate warming and losses of a valuable energy resource. The oil and gas value chain contributes at least 25% of anthropogenic methane emissions globally and is the second largest methane-emitting sector in the United States. Here, we assess methane
emissions in regions accounting for 70% of US onshore oil and gas production in 2023 using data collected by the MethaneAIR airborne imaging spectrometer. We quantify total methane emissions across all observed regions to be ~9 (7.8 – 10) Tg/yr, with ~90% of emissions estimated from the oil and gas sector (~8 Tg/yr, equivalent to a methane loss rate of 1.6% of gross gas production), which is about five times higher than reported by the US EPA. Both oil and gas emissions and gas production-normalized methane loss rates varied considerably by basin. Highly productive basins such as the Permian, Appalachian, and
Haynesville-Bossier had the highest methane emissions (95 – 314 t/hr), whereas lower producing basins possibly associated with older infrastructure such as the Uinta and Piceance had higher loss rates (>7%). We found good agreement across total emissions quantified by MethaneAIR and other empirical and remote sensing estimates at national/basin/target-level scales. This work underscores the increasing value of remote sensing data for quantifying methane emissions, characterizing intensity of methane losses across the oil and gas sector, and mapping inter-basin emissions variability, which are all critical for tracking
methane mitigation targets set by industry and governments.

## 1 Introduction

Over 150 countries and 50 oil and gas companies have pledged to substantially reduce their methane emissions in this decade in efforts to combat climate change (The Oil and Gas Decarbonization Charter, 2024; Global Methane Pledge, 2024). Methane is a short-lived (atmospheric lifetime of 9 – 11 years) and potent greenhouse gas and its reduction can substantially slow the





rate of climate warming in the near term, which is critically needed to avoid the worst effects of anthropogenic climate change (Ocko et al., 2021).

Significant technical advances in methane measurement have been made in recent years, including new remote sensing technologies now being deployed at scale, with the oil and gas sector being the primary focus of such measurements (Jacob et

al., 2022; Zhang et al., 2023). Recent measurements have revealed important information on the sources and magnitudes of oil and gas methane emissions, with several studies concluding that industry and governments who rely on bottom-up estimation are underreporting methane emissions (Alvarez et al., 2018; MacKay et al., 2021; Omara et al., 2024; Shen et al., 2022; Sherwin et al., 2024; Stavropoulou et al., 2023; Zavala-Araiza et al., 2021), and in some cases are inaccurately estimating the relative contributions of different sources (Conrad et al., 2023a, b). Direct measurements of emissions play a fundamental role in

methane reduction efforts by helping to reduce uncertainties in "bottom-up" source-level inventories needed to inform efficient mitigation and to track reductions over time.

MethaneSAT is a satellite mission (launched on March 4, 2024) designed to provide quantitative data on total regional methane emissions, with a goal of mapping emissions in regions accounting for over 80% of global oil and gas production

(MethaneSAT, 2024). MethaneAIR is an airborne precursor instrument with similar spectroscopy to MethaneSAT (Chan Miller et al., 2023; Staebell et al., 2021). In 2023, MethaneAIR was flown on a modified Lear 35 jet operating at about 12 km altitude to map methane emissions from major oil and gas producing regions in the United States. Compared to ground-based measurement techniques, high altitude aerial systems like MethaneAIR can cover much larger areas in less time, and are not limited by site accessibility, making them particularly useful for assessing and comparing methane emissions at the basin-

level. MethaneAIR was designed to detect and quantify both area aggregates of dispersed emission sources as well as high-emitting point sources, measuring total regional emissions with high precision and spatial resolution.

In this study, we analyze MethaneAIR data from over 30 flights conducted from June to October 2023, covering 12 oil and gas basins that account for 70% of contiguous United States (CONUS) onshore oil and gas production in 2023. We use this

data to quantify and assess basin-level methane emissions, as well as compare total methane emissions and gross gas production-normalized loss rates across oil and gas production basins. As part of the analysis, we also provide estimates of the oil/gas fraction of total emissions for individual observed regions based on MethaneAIR data in combination with an in depth assessment of previously published estimates. Finally, we compare total methane emissions quantified by MethaneAIR to other independent measurements and empirical data available in recent peer-reviewed literature, and to the Environmental

Protection Agency's Greenhouse Gas Inventory (EPA GHGI) (Maasakkers et al., 2023).



## 2 Methods

### 2.1 Overview of measurement campaign

In 2023, MethaneAIR collected measurements over 12 oil and gas basins across the US. Basins were selected based on their production levels and characteristics, such that measured regions covered the majority of US onshore oil and gas production

and the diverse range of basin characteristics within the country (e.g., mixed production, oil-dominant, gas-dominant, mature, rapidly developing). A brief overview of each basin covered in this study and its characteristics is in Table 1. Figure 1 shows the MethaneAIR flight domains within each basin, colored by the month in which each flight occurred. Combined, these measured areas account for 70% of CONUS onshore oil and gas production in 2023.

**Table 1: Summary statistics for the 12 oil and gas basins covered in this study. 2023 annual production data is from Enverus (Enverus: Prism, 2024), expressed as million barrels of oil equivalent (Mboe), using a conversion factor of 1 boe = 6,000 cubic feet of natural gas and 1 boe = 0.14 toe. Estimated primary sources of methane emissions are based on the EPA GHGI for 2020 (Maasakkers et al., 2023). Note that minor sources (contributing less than 10% of total) are not listed, therefore percentages do not always add up to 100%.**

| Basin | Basin area (km²) | 2023 oil production (Mboe) | 2023 gas production (Mboe) | Percent of total production from oil/gas | Estimated primary sources of methane emissions |
|---|---|---|---|---|---|
| Anadarko | 42,500 | 100 | 342 | 23% / 77% | Oil & gas (77%), agriculture (20%) |
| Appalachian | 415,000 | 63 | 2,167 | 3% / 97% | Oil & gas (39%), coal (32%), agriculture (16%) |
| Arkoma-Fayetteville | 15,200 | 0.3 | 63 | 1% / 99% | Oil & gas (70%), agriculture (21%) |
| Bakken | 67,200 | 442 | 200 | 69% / 31% | Oil & gas (72%), agriculture (21%) |
| Barnett | 68,100 | 7 | 144 | 4% / 96% | Oil & gas (62%), agriculture (26%), waste (12%) |
| Denver-Julesburg | 33,700 | 170 | 183 | 48% / 52% | Oil & gas (40%), agriculture (36%), waste (23%) |
| Eagle Ford | 50,200 | 432 | 487 | 47% / 53% | Oil & gas (75%), agriculture (21%) |
| Greater Green River | 67,000 | 9 | 147 | 6% / 94% | Oil & gas (64%), coal (24%) |
| Haynesville-Bossier | 28,900 | 8 | 983 | 1% / 99% | Oil & gas (79%), waste (11%) |
| Permian | 165,000 | 2,162 | 1,410 | 61% / 39% | Oil & gas (86%), agriculture (11%) |
| Piceance | 34,500 | 5 | 73 | 6% / 94% | Oil & gas (66%), coal (19%), agriculture (10%) |
| Uinta | 37,800 | 52 | 46 | 53% / 47% | Oil & gas (63%), coal (22%), agriculture (13%) |




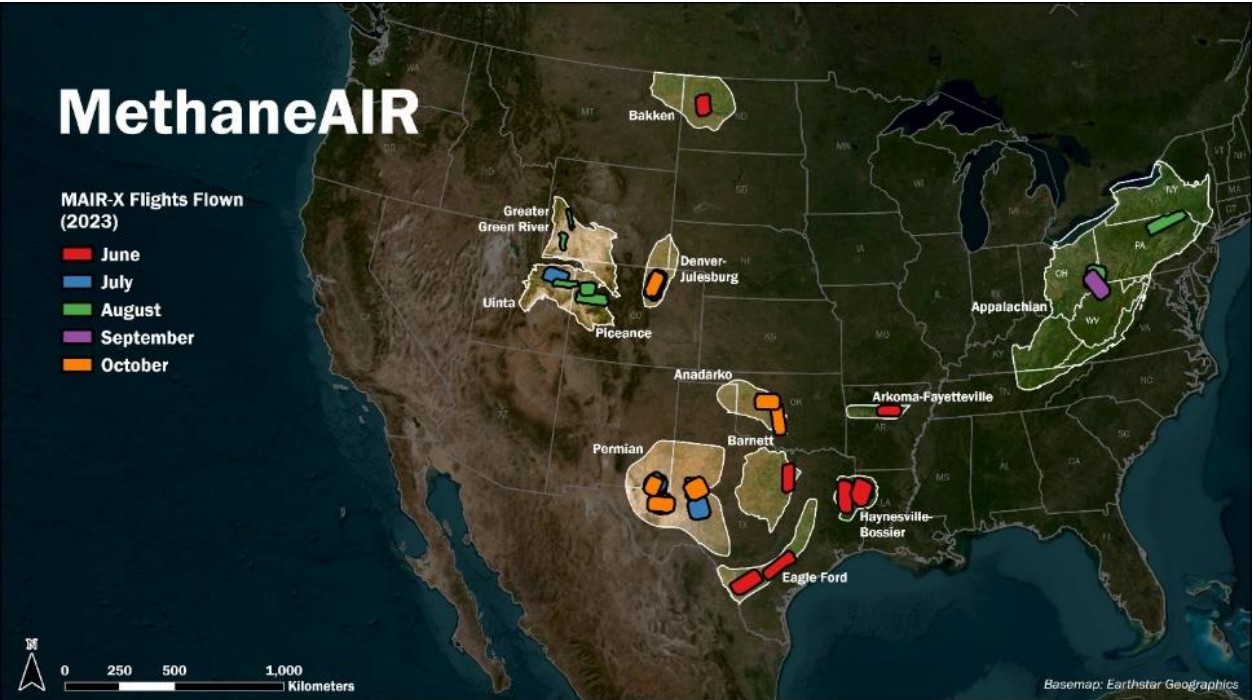

**Figure 1: MethaneAIR flight domains in each oil and gas basin covered in this study, colored according to the month each flight occurred. Basin boundaries are outlined in white. Measured regions cover areas with high oil and gas production.**

The MethaneAIR Lear 35 jet operated at about 12 km altitude, with each flight covering approximately 10,000 km$^2$ over two

hours. The MethaneAIR technical specifications, calibration, data processing and validation have been described in recent studies by Chan Miller et al., 2023; Chulakadabba et al., 2023; Conway et al., 2024; El Abbadi et al., 2024; Omara et al., 2024; Staebell et al., 2021; Warren et al., 2024. Discussion of the quantification methods are provided in SI Sect. S1 and briefly summarized here.

Total regional methane emissions for each MethaneAIR flight were quantified using an inverse model that finds the gridded emission rates that best explain observed column-averaged dry-air mole fractions of methane. Emissions and observed methane concentrations are linked by a Jacobian matrix computed using the Stochastic Time-Inverted Lagrangian Transport (STILT) model (Fasoli et al., 2018; Lin et al., 2003). The inversion framework utilizes the instrument's high spatial resolution, wide spatial coverage, and high precision. High-emitting (> ~200 kg/hr) discrete point sources are quantified in a preliminary

analysis using a divergence integral method (Abbadi et al., 2023; Chulakadabba et al., 2023; Warren et al., 2024), and their associated methane enhancements are computed by propagating through the Jacobian, which are then subtracted from the observations. This procedure places trust in the well-tested point-source specific algorithm to quantify high-emitting point sources and uses the Jacobian to ensure the complete mass of methane from point sources is accounted for, without double counting. The resulting analysis of MethaneAIR data produces a high resolution (1 km by 1 km), spatially explicit



quantification of methane emissions, as well as the specific location and quantification of individual point sources emitting above ~200 kg/hr. The total emission rate for the region is obtained by summing the dispersed area sources and the point source emissions. More information about the emission quantification approach is provided in SI Sect. S1.

Like many other remote sensing technologies, an important caveat to note about MethaneAIR measurements is that they are
collected over a relatively short time (over two hours for the measurement domain), and only during the day. Therefore, methane emissions estimated from measurements collected during a single flight may not adequately capture potential variability in emissions that occur throughout a 24-hour period, or longer (e.g., weeks, months).

## 2.2 Previously published measurement-based data

This study leverages previously published measurement-based methane emissions estimates from several studies summarized
in Table 2, which can be referred to directly for more in-depth descriptions of their respective methodologies. Data from these studies were used to develop independent measurement-based estimates and ranges of methane emissions within MethaneAIR spatial domains for intercomparisons and to inform estimates of the relative contributions of oil and gas sources relative to total emissions quantified by MethaneAIR. We used the following criteria to determine which studies to include: 1) geographic overlap with MethaneAIR spatial domains, 2) methane emissions are reported either as spatially explicit (i.e., gridded)
estimates, or as regional totals for a defined domain, and 3) recent measurements in the peer-reviewed literature with a majority of studies included that were collected between 2019 – 2021.

Where studies provide gridded methane emissions data products (e.g., Lu et al., 2023; Shen et al., 2022), we extracted and computed total methane emissions for the MethaneAIR study domains. For studies that report only total methane emissions
for a defined area (Lin et al., 2021; Sherwin et al., 2024), we first compared the study domain with the MethaneAIR domains to assess the relative overlap. If there is a >50% geographic overlap with a given MethaneAIR domain, then the study is included in subsequent comparisons and analysis.

**Table 2: Summary of previous measurement-based data included in this work.**

| Study | Measurement platform | Methodological details | Spatial Coverage | Source Coverage | Study period |
|---|---|---|---|---|---|
| Alvarez et al., 2018 | Various ground-based and aerial | Synthesis of previous measurements | Haynesville, Barnett, Appalachian, San Juan, Fayetteville, Bakken, Uinta, Arkoma, Denver-Julesburg | Oil and gas | 2015 |
| Barkley et al., 2017 | Aerial | Mass balance and inverse modelling (WRF-Chem) | Appalachian (NE PA) | All | 2015 |





| Barkley et al., 2023 | Stationary towers | Continuous tower measurements, inversion with prior | Appalachian (NE PA), Permian (Delaware) | All | 2015 – 2016 |
|---|---|---|---|---|---|
| Cusworth et al., 2022 | Aerial and satellite | CarbonMapper and TROPOMI (no prior) | Denver-Julesburg, Permian, Appalachian, San Juan, Uinta | All | 2019 – 2021 |
| Fried and Dickerson, 2023 | Aerial | Mass balance | Denver-Julesburg | All | 2021 |
| Lin et al., 2021 | Stationary towers | Langrangian Particle Dispersion Modeling technique (STILT) | Uinta | All | 2020 |
| Lu et al., 2023 | Satellite | GOSAT & surface observations (continental-scale GEOS-Chem chemical transport model; with prior) | National | All | 2019 |
| Nesser et al., 2024 | Satellite | TROPOMI (with prior) | National | All | 2019 |
| Omara et al., 2024 | Various ground-based | Facility-level measurement-based inventory (EI-ME) | National | Oil and gas | 2021 |
| Peischl et al., 2018 | Aerial | Mass balance (SONGNEX NOAA P-3) | Denver-Julesburg, Bakken, Barnett, Eagle Ford, Haynesville | All | 2015 |
| Schwietzke et al., 2017 | Aerial | Mass balance | Arkoma-Fayetteville | All | 2015 |
| Shen et al., 2022 | Satellite | TROPOMI (with prior) | National | Oil and gas | 2018 – 2020 |
| Sherwin et al., 2024 | Aerial | Carbon Mapper (AVIRIS-NG) and bottom-up simulations | Permian, San Joaquin, Denver-Julesburg, Appalachian, Uinta | Oil and gas | 2020 |
| Varon et al., 2023 | Satellite | TROPOMI (weekly inversions with previous week as prior) | Permian | All | 2018 – 2020 |
| Veefkind et al., 2023 | Satellite | TROPOMI (no prior) | Permian | All | 2019 – 2020 |
| Zhang et al., 2020 | Satellite | TROPOMI (with prior) | Permian | All | 2018 – 2019 |

**2.3 Analysis**

**2.3.1 Estimating ranges of total methane emissions for study regions using previous data**

For each MethaneAIR flight domain, we perform bootstrap resampling with replacement (n = 5,000) of previous estimates selected from the studies listed in Table 2 to develop ranges (i.e., mean and 95% confidence intervals) of oil and gas, non-oil and gas, and total methane emissions for the area. Briefly, we define oil and gas emissions as those originating from activities

and infrastructure involved in the production, processing, transport, and distribution of oil and natural gas (e.g., well sites, pipelines, compressor stations, natural gas processing plants). Non-oil and gas emissions are any emissions from coal,



agricultural (e.g., livestock, manure management), waste (e.g., landfills, wastewater treatment), and other industrial sectors. For domains with less than four unique estimates, we use the minimum and maximum estimates from previous studies as the range. In some cases, bottom-up data (e.g., Crippa et al., 2024; Maasakkers et al., 2023) are used to inform the ranges of non-

oil and gas methane emissions due to a lack of measurement-based data (see SI Sect. S2 for additional discussion). We further use the ranges for non-oil and gas methane emissions to estimate the relative sector (i.e., oil and gas, non-oil and gas) contributions of total emissions quantified by MethaneAIR.

### 2.3.2 Estimating sector contributions and gross gas-normalized methane loss rates using MethaneAIR data

We estimate total oil and gas methane emissions (Eq. 1) as the difference between the total methane emissions quantified by

MethaneAIR, $MethaneAIR_{total}$, and an estimate of total non-oil and gas methane emissions in the MethaneAIR observation domain based on non-oil and gas methane emission estimates from previous literature, $LEmis_{non-ong}$. We also subtract any point source emissions quantified by MethaneAIR attributed to non-oil and gas sources, $Pt\_MAIR_{non-ong})$, which were observed in five out of 12 basins.

$$MethaneAIR_{ong} = MethaneAIR_{total} - (LEmis_{non-ong} + Pt_{MAIR_{non-ong}}) \quad\quad\quad\quad (1)$$

For the above method, we acknowledge that subtracting a literature-based estimate of non-oil and gas methane emissions as well as non-oil and gas point source emissions quantified by MethaneAIR may introduce double counting as some of the MethaneAIR point source emissions may also be captured in literature-based estimates. However, since bottom-up data is included in many of the literature-based estimates for these regions, it is possible that the estimate could be low-biased, since recent research has shown that methane emissions from sectors such as waste are being under reported in bottom-up inventories

(Cusworth et al., 2024; Moore et al., 2023; Nesser et al., 2024). If this is the case, the potential issue of double counting could be negligible, although difficult to confirm with current limited empirical data on non-oil and gas emissions. To help us understand whether this approach is reasonable, we also explore another method to estimate the relative sector contributions of methane emissions for regions of interest. Our alternative method utilizes spatially explicit methane emissions data for oil and gas sources from the measurement based inventory developed by Omara et al., 2024, updated using 2023 activity data,

and non-oil and gas sources from the EPA 2020 GHGI (Maasakkers et al., 2023) to estimate grid-level ratios of oil and gas emissions, which are then applied to the quantified area emissions from MethaneAIR retrievals. This additional approach is further discussed in SI Sect. S2, with detailed comparisons of estimates derived by the two methods shown in Fig. S2.

MethaneAIR data has sufficient spatial resolution and precision to disaggregate emissions by sector in many basins, and

sufficient swath to determine regional totals. Robust, observation-based disaggregation of methane emissions across sectors is an important part of an actionable and policy relevant data analysis, but it is challenging in some regions due to the commingling of different sectors geographically, and because of limited empirical data on non-oil and gas methane emissions.





The empirically based sector disaggregation presented here is a first step. Future work will continue to explore and refine these methods as both still have uncertainties that need to be better assessed.


Gross gas production normalized oil and gas methane loss rates, expressed here as the percentage of methane emitted relative to total methane produced (Eq. 2), are calculated by dividing the total oil and gas methane emissions estimated from MethaneAIR data by the gross methane produced in the measured regions. Gross methane production is estimated using 2023 gross natural gas production data (Enverus: Prism, 2024) and basin-specific gas compositions (Table S2) that are consistent

with previous literature assumptions on methane composition.

$$Methane\ loss\ rate\ (\%) = \frac{Oil\ and\ gas\ methane\ emissions\ (\frac{kg}{hr})}{(Gross\ natural\ gas\ production\ (\frac{kg}{hr}) \times methane\ content)} \times 100 \qquad (2)$$

As an additional metric for comparisons, we also computed energy-normalized methane intensities (in kg $CH_4$/GJ) by dividing oil and gas methane emissions estimated from MethaneAIR by the combined gross oil and gas production (Enverus: Prism, 2024) in the measured regions, similar to intensities reported by the International Energy Agency for their annual Global

Methane Tracker (IEA, 2025). See SI Sect.S5 for additional discussion and comparisons of both metrics across all measured basins.

### 2.3.3 Basin-level aggregation of MethaneAIR data

We spatially aggregated overlapping MethaneAIR flights in the 12 measured oil and gas basins to produce an estimate of total methane emissions and associated uncertainties for each basin. Two separate approaches for aggregation were explored, which

we defined as 1) unique overflown area (UOA) averaging, and 2) area-normalized averaging, with both methods producing similar results (Fig. S4). Results from the UOA averaging method are presented in the main text, and comparisons and additional discussion on the other method can be found in SI Sect. S3.

For the UOA averaging method, we first mapped the spatial domains of each MethaneAIR flight to identify areas that were

uniquely overflown by the same combination of flights, which creates a subset of smaller spatial domains (Fig. S3). Next, we iterated through the subset of smaller spatial domains (i.e., denoted as UOA, or unique overflown areas) and averaged both the point source and area emissions quantified from the corresponding flights. The resulting averages of all UOAs are then summed to produce a total estimate of methane emissions for the aggregated flight domains at the basin-level.

To calculate basin-level uncertainties in the dispersed area emissions using the UOA approach, we first adjust the uncertainties for the dispersed emissions for each UOA based on the percentage of area covered using Eq. 3, where $U_f$ is the uncertainty in the dispersed area emissions at the flight level, $A_f$ is the area covered by the entire flight, and $A_{UOA}$ is the area covered by the UOA which is a subset of the entire flight domain. This adjustment accounts for the inherently higher uncertainties contained within spatial subsets of the entire flight domain, with the assumption that uncertainties are uniform across the domain.



Additional refinement to the uncertainties in the dispersed area emissions should incorporate parameters such as the effects of albedo, terrain, and weather conditions to produce more accurate estimates of the uncertainties across different portions of the spatial domain for a given flight. A comparison of basin-level uncertainties with and without the area-based adjustment show very minimal differences (1 – 3%), except in the Permian and Denver-Julesburg basins where the area-adjusted approach increases the uncertainties by >2x due to more unique flights occurring in these basins. That said, the area-adjusted approach is possibly a more conservative method for estimating uncertainties in these two basins.

$$U = \sqrt{\frac{A_f}{A_{UOA}}} U_f \tag{3}$$

Next, using the adjusted dispersed area emissions uncertainties (Eq. 3) we propagate the uncertainties for all flights corresponding to a UOA (Eq. 4) where $U_i$ is the area uncertainty from flight $i$ for a UOA, $n$ is the total number of flights within the UOA, and $U_{UOA}$ is the uncertainty for the UOA.

$$U_{UOA} = \frac{\sqrt{U_i^2 + \cdots + U_n^2}}{n} \tag{4}$$

To calculate the uncertainties of point sources for each UOA, we resampled point sources (n = 5,000) from flights within a UOA assuming the uncertainty for a single point source follows a normal distribution with parameters of the distribution as the point source standard deviation and the mean as the quantified emission rate. The 2.5th and 97.5th percentiles from the resulting distribution are used as the associated uncertainties. Then, we used Eq. 4 to determine the average point source uncertainties for each UOA.

The overall uncertainties for aggregated basin-level total emissions are estimated using Eq. 5 for the area emissions and point source emissions separately, where $U_{total}$ is the aggregated basin-level uncertainty, $U_i$ is the percentage uncertainty for an emission estimate from a UOA within the basin, and $x_i$ is the associated emission rate for the UOA. Finally, Eq. 5 is used again to combine point source and area emissions uncertainties to produce the overall uncertainty bounds for the aggregated basin-level and national-level total emissions estimates. Uncertainties on the basin-level oil and gas estimates and associated loss rates incorporate the quantification uncertainty as described above (for both area and point source quantification), and an estimate of uncertainty related to the subtraction of literature-based estimates of non-oil and gas emissions (Eq. 1) using the standard deviation of the bootstrapped distribution.

$$U_{total} = \frac{\sqrt{(U_i \cdot x_i)^2 + \cdots + (U_n \cdot x_n)^2}}{|x_i + \cdots + x_n|} \tag{5}$$

It is important to note that the resulting estimates represent the total methane emissions for the area within each basin that was explicitly measured by MethaneAIR across multiple flights, and not the entire geographic extent of each basin (Fig. 1). However, these measured areas cover more than two-thirds of each basin's total oil and gas production with several exceeding 90% of the basin's production (except for the Bakken and Greater Green River basins) and combined make up 70% of the CONUS onshore oil and gas production in 2023 (Table 3).





## 3 Results and discussion

### 3.1 Basin by basin comparison of MethaneAIR quantification

#### 3.1.1 Comparison of total methane emissions

Figure 2 and Table 3 show total methane emissions estimated by aggregating MethaneAIR data collected in 12 major oil and
gas producing basins, delineated by estimated relative contributions of oil and gas and non-oil and gas sources. When considering methane emissions from all sectors, the Permian, Appalachian, and Haynesville-Bossier basins rank highest in terms of absolute methane emissions. In addition to oil and gas methane emissions, the Appalachian has significant emissions from the coal and waste sectors, which was also observed in previous work (Barkley et al., 2019; Cusworth et al., 2022). We estimate that around 14% of emissions in the Haynesville-Bossier basin are from non-oil and gas methane emissions, most of
which are likely from the waste sector (Maasakkers et al., 2023), and for the Permian, we found that almost all (>95%) emissions are from oil and gas activity (Fig. 2).

The Barnett and Denver-Julesburg basins are additional regions with larger contributions of non-oil and gas methane emissions (>35%), which we attribute primarily to the agriculture and waste sectors (Crippa et al., 2024; Cusworth et al., 2022; Lu et al.,
2023; Maasakkers et al., 2023; Peischl et al., 2018). However, based on our analysis, we estimate that oil and gas methane emissions make up the majority of methane emissions in all 12 basins, ranging from 57 – 99% of the total. As discussed in Sect. 2.3.2, these percent contributions have varying levels of uncertainty due to limited data.

It is important to note that estimates for some basins are based on a single MethaneAIR flight (Table 3), whereas others are
based on several flights occurring over the span of weeks or months. Repeat overpasses throughout the year are needed to produce a more representative estimate of basin-level methane emissions.



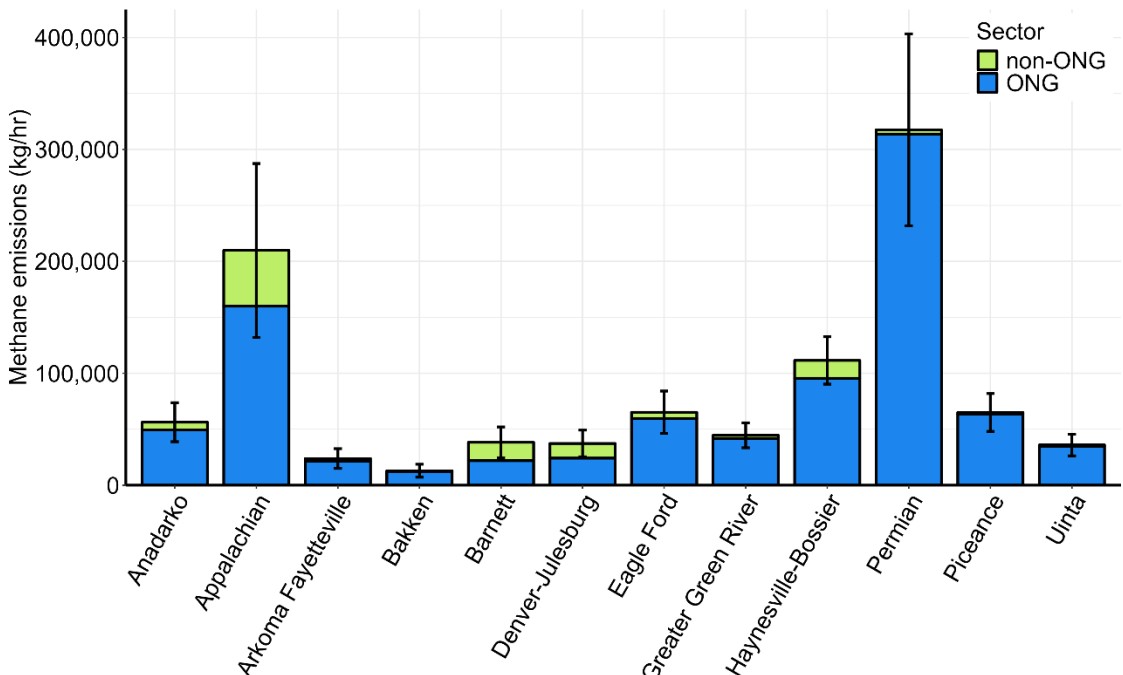

**Figure 2: Basin-level estimates of total methane emissions quantified by MethaneAIR with estimated contributions from oil and gas and non-oil and gas sectors. Total emissions across measured regions vary by an order of magnitude, with the Permian basin having the highest total emissions. Non-oil and gas emissions are most prevalent in the Appalachian, Barnett, Haynesville-Bossier, and Denver-Julesburg basins.**

### 3.1.2 Comparison of oil and gas methane emissions and loss rates

Figure 3 and Table 3 show the estimated total oil and gas methane emissions and gas normalized loss rates over individual oil/gas basins (following methods discussed in Sect. 2.3.2). When considering only oil and gas related methane emissions, the Permian, Appalachian, and Haynesville-Bossier remain as the highest emitting basins. These basins are also the top oil and gas producers out of all basins included in this study (Table 1). The Permian is dominated by oil production but has significant associated gas production, accompanied by increasing new oil and gas development and high amounts of flaring, all of which could lead to higher observed methane emissions (Lyon et al., 2021; Varon et al., 2023; Zhang et al., 2020).

While absolute emissions reveal important insights as noted above, methane loss rates are an important metric to consider when comparing methane performance across basins with variable levels of gas production. For instance, gas-dominant basins with high well-site productivity (Appalachian, Haynesville-Bassier) have the lowest methane loss rates (<1%), despite having some of the highest absolute emissions (Fig. 3, Table 3). Oil-dominant or mixed oil/gas basins (e.g., Permian, Greater Green River, Eagle Ford, Bakken) tend to have higher methane loss rates (2 – 5%). Relatively mature basins where oil and gas production and well site infrastructure is dominated by large populations of aging, low producing wells such as the Piceance





and Uinta have the highest observed methane loss rates (>7%) (Fig. 3, Table 3), likely due to fugitive methane emissions that continue to occur even as production declines (Lin et al., 2021; Omara et al., 2022). We also estimated energy-normalized methane intensities (kg $CH_4$/GJ) for each basin to compare oil and gas methane emitted relative to each basin's combined oil and gas production, which are discussed in SI Sect. S5.


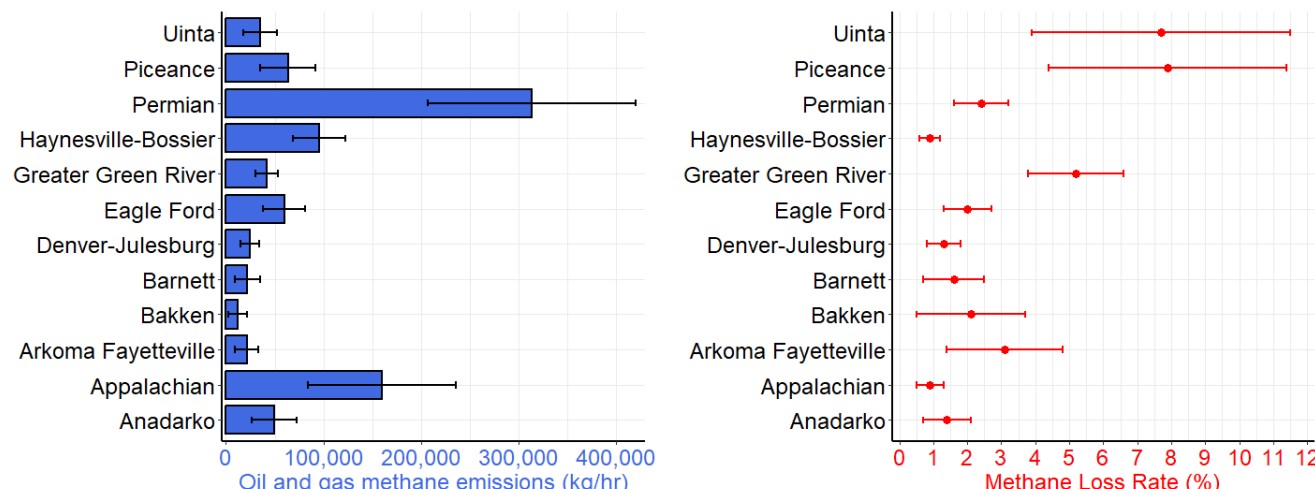

**Figure 3: Basin-level oil and gas methane emissions (kg/hr) and gas normalized loss rates (%) estimated from MethaneAIR data. Gas-dominant basins (Appalachian and Haynesville-Bossier) have the lowest loss rates (<1%), whereas low-producing basins with aging infrastructure (Piceance and Uinta) have higher loss rates (>7%).**

**Table 3: MethaneAIR estimated total methane emissions, oil and gas methane emissions, and gross gas normalized methane loss rates for measured regions within each basin. Production coverage is based on 2023 Enverus data (Enverus: Prism, 2024).**

| Basin | Number of unique MethaneAIR flights | Percent of basins total oil and gas production covered by MethaneAIR | MethaneAIR total methane emissions (kg/hr) and uncertainty | MethaneAIR estimated oil and gas methane emissions (kg/hr) and uncertainty | MethaneAIR gas-normalized methane loss rates (%) |
|---|---|---|---|---|---|
| Anadarko | 2 | 78% | 56,300 (±31%) | 49,500 (±46%) | 1.4% |
| Appalachian | 3 | 68% | 209,800 (±37%) | 160,000 (±47%) | 0.9% |
| Arkoma Fayetteville | 1 | 88% | 23,800 (±37%) | 21,300 (±57%) | 3.1% |
| Bakken | 1 | 48% | 12,900 (±45%) | 12,300 (±78%) | 2.1% |
| Barnett | 1 | 80% | 38,300 (±36%) | 21,900 (±60%) | 1.6% |
| Denver-Julesburg | 5 | 91% | 37,300 (±32%) | 24,200 (±40%) | 1.3% |
| Eagle Ford | 2 | 73% | 65,200 (±29%) | 59,600 (±36%) | 2.0% |
| Greater Green River | 2 | 44% | 44,600 (±25%) | 41,800 (±27%) | 5.2% |
| Haynesville-Bossier | 3 | 95% | 111,000 (±19%) | 95,400 (±28%) | 0.9% |





| | | | | | |
|---|---|---|---|---|---|
| Permian | 8 | 88% | 318,000 (±27%) | 314,000 (±34%) | 2.4% |
| Piceance | 2 | 93% | 65,000 (±26%) | 63,500 (±44%) | 7.9% |
| Uinta | 2 | 92% | 35,900 (±27%) | 34,900 (±50%) | 7.7% |
| **12 basin sum** | **32** | **-** | **1,018,000 (±12%)** | **898,000 (±16%)** | **1.6%** |

### 3.1.3 Comparison of MethaneAIR estimated emissions to the EPA GHGI

Figure 4 (A) shows basin-level estimates of total methane emissions quantified by MethaneAIR compared to total emissions reported by the EPA (Maasakkers et al., 2023) for the same domains. Note that EPA estimates are for 2020 (the most recent

year available at the time of writing), and MethaneAIR measurements were collected in 2023. Depending on the basin, MethaneAIR estimates of total methane emissions range from 1.8 (Barnett) to 8.2 (Greater Green River) times higher than EPA estimates. While it is possible that some of these differences may be due to actual changes in emissions between 2020 and 2023 (e.g., from changes in activity), it is unlikely that such changes would result in the large discrepancies observed, suggesting that underreporting of emissions remains an issue for these regions.


Basin-level oil and gas methane emissions estimated by MethaneAIR similarly range from 1.3 (Barnett) to 7.9 (Greater Green River) times higher than EPA estimates (Fig. 4 B), suggesting that observed discrepancies can be primarily attributed to the oil and gas sector. Despite several previous measurement studies finding similar differences between measured and reported emissions (Alvarez et al., 2018; Omara et al., 2024; Sherwin et al., 2024; Zhang et al., 2020), some of which date back to over

a decade ago (Brandt et al., 2014), our analysis indicates that underreporting continues to be prevalent for major oil and gas producing basins in the US, which must be addressed if such inventories are to be used to inform mitigation and track promised reductions over time.

To better contextualize this comparison, we assess the relative contributions of EPA's reported oil and gas methane emissions

from the MethaneAIR measurement domains compared to their reported emissions for the rest of the onshore CONUS. EPA estimates only 21% (1.6 Tg/yr, 0.4% loss rate) of the total CONUS onshore oil and gas methane emissions are from sources within the MethaneAIR domains, which are responsible for more than 70% of total onshore oil and gas production in the CONUS for 2023. This suggests that EPA estimates the other 79% of the onshore CONUS oil and gas methane emissions (6.1 Tg/yr, 3.6% loss rate) are from the regions outside of the MethaneAIR domains, which make up only 30% of total onshore oil

and gas production for the CONUS in 2023. Considering these findings, it is possible that the discrepancies between EPA and MethaneAIR are partly attributable to EPA's use of methane emission factors (e.g., estimate of average methane emitted per gas well) that may be unrepresentative across basins.





**Figure 4: Total methane emissions (A) and estimates oil and gas methane emissions (B) quantified by MethaneAIR compared to EPA (Maasakkers et al., 2023) reported emissions for the same regions. MethaneAIR estimates (blue bars) of total methane emissions range from 1.8 (Barnett) to 7.9 (Greater Green River) times higher than EPA estimates (grey bars).**



## 3.2 Methane emissions quantified by MethaneAIR from 70% of US onshore production

We estimate total methane emissions across all measured regions are 8.9 (7.8 – 10) Tg/yr (assuming emissions are constant throughout the year), with ~90% (7.9 Tg/yr) of emissions coming from the oil and gas sector (Table 3). Comparing the

combined methane emissions from all measured regions within the 12 oil and gas basins, the MethaneAIR total is approximately four times higher than total emissions reported by EPA. The MethaneAIR oil and gas total for all basins corresponds to a methane loss rate of 1.6% (or a methane intensity of 0.17 kg $CH_4$/GJ), which is more than four times higher than EPA's loss rate (0.4%) for the same regions, and more than eight times higher than the intensity target in the Oil and Gas Decarbonization Charter (The Oil and Gas Decarbonization Charter, 2024). The observed differences between reported and

measured gas-normalized methane loss rates are similar to those previously reported for the US (Alvarez et al., 2018; Omara et al., 2024; Sherwin et al., 2024). The estimated energy-normalized methane intensity of 0.17 kg $CH_4$/GJ is comparable to the upstream methane intensity of 0.18 kg $CH_4$/GJ for the entire US reported by the IEA for 2024 (IEA, 2025), however their estimate is calculated using marketed oil and gas production, whereas our estimate uses gross production which inherently results in lower intensities.

## 315   3.3 Regional (i.e., flight-level) comparisons of total methane emissions and loss rates

We compared the MethaneAIR quantified methane emissions and estimated gross gas normalized methane loss rates from individual flights to other measurement-based estimates from independent ground-based, aerial, and satellite platforms. Figures 5 and 6 show these comparisons for six flights in the Haynesville-Bossier, Barnett, Eagle Ford, Permian, Denver-Julesburg, and Anadarko basins (additional comparisons for other flights are in SI Sect. S7). Note that in Figure 6, some previous

measurement-based estimates include only oil and gas methane emissions (dark blue bars), whereas the MethaneAIR estimates and others with light blue bars are total methane emissions (from all sectors). Across these flights, as well as the majority of the other 26 included in the present analysis, the MethaneAIR quantification generally shows good agreement with previous measurement-based estimates. Minor differences observed across independent measurements could be due to several factors such as the differences in time and duration of measurements as well as likely variability in emissions over time.


The broad agreement between methane emissions quantified by MethaneAIR and other independent measurements further builds confidence in MethaneAIR's capability to provide robust quantification of methane emissions over large areas. Similarly, the agreement between our MethaneAIR-based loss rate estimates and other independent measurement-based estimates builds confidence in our methods for assessing the contributions of the oil and gas sector to total methane emissions.





**Figure 5: Total methane emissions for six MethaneAIR flights compared to other measurement-based estimates reported in the literature. The grey shaded area and dashed lines show the representative ranges of total methane emissions (95% CI, minimum/maximum) derived from previous literature. The MethaneAIR quantification across different measured basins shows good agreement compared to other measurement-based estimates.**




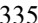

**Figure 6: Gross-gas production normalized methane loss rates for six MethaneAIR flights compared to other measurement-based estimates reported in the literature. The grey shaded area and dashed lines show the representative ranges of loss rates (95% CI,**



**minimum/maximum) derived from previous literature. The MethaneAIR estimated loss rates across different measured basins show**
**relatively good agreement compared to other loss rate estimates.**

## 4 Conclusions

We used MethaneAIR data from 32 flights to quantify and compare methane emissions across 12 oil and gas production basins in the US that account for 70% of national onshore oil and gas production. Our results suggest that these regions emit approximately 8.9 Tg/yr of methane, with ~10% of emissions from non-oil and gas sources (e.g., coal, landfills, and

agriculture). Oil and gas methane emissions and gross gas production-normalized loss rates estimated for individual basins vary significantly, likely due to a combination of differences in production, infrastructure, and operational practices. Because of these variations, effective mitigation strategies may need to be tailored for individual basins. Additionally, we found that some of the highest emitting basins have lower loss rates, and vice versa, highlighting the importance of considering both metrics when evaluating the methane performance of a particular basin or region. More data in terms of repeat and systematic

surveys throughout the year are needed to further characterize the inter-basin emissions variability.

We found good agreement in methane emissions characterized by MethaneAIR and other independent measurement-based estimates, adding confidence in the capability of MethaneAIR data to quantify total regional methane emissions. Similar to previous studies, we found observed emissions to be much higher than what is currently reported in bottom-up inventories.

Emission quantification provided by MethaneAIR data, along with other empirical and remote sensing data, can be used to address gaps and improve estimates in existing bottom-up inventories to more accurately track progress towards methane mitigation targets set by industry and governments. Some countries such as Canada have started to incorporate atmospheric measurements in their official inventories (Environment and Climate Change Canada, 2024), which has significantly reduced the gap between measured and reported oil and gas methane emissions (MacKay et al., 2024).


With regards to sector-disaggregation, we applied various approaches and discussed challenges related to estimating sector-specific methane emissions from total regional emissions in areas with multiple methane emitting sectors. Developing robust assessments of sector contributions is essential for providing actionable and policy-relevant insights from remote sensing measurements. More sector-specific empirical data are needed to further characterize oil/gas and non-oil/gas emissions

disaggregation by employing facility-level measurements and modeling. In our analysis, the Appalachian, Denver-Julesburg, and Barnett basins were identified as having relatively significant contributions of non-oil and gas methane emissions and would be regions that would especially benefit from future research on this topic.



**Data availability**

MethaneAIR L3 and L4 data are available for download via Google Earth Engine (https://developers.google.com/earth-engine/datasets/tags/methanesat). Emissions data from other measurement-based studies are available online, please refer to the original publications referenced in Table 2 for access information. The EPA gridded GHGI data is available at https://www.epa.gov/ghgemissions/us-gridded-methane-emissions. EDGARv8 data is available at https://edgar.jrc.ec.europa.eu/dataset_ghg80. Basin boundaries are based on US EIA basin boundaries data (https://www.eia.gov/maps/maps.php).

**Author contributions**

Conception and design of this study was led by RG, SW. Analysis and interpretation of the data was done by KM, JB, JPW, MO, AH, MS, JDW, CCM, SR, ZZ, and LG. KM drafted and revised the manuscript with contributions from all authors.

**Competing interests**

The corresponding authors declare that none of the authors has any competing interests.

**Acknowledgements**

Funding for MethaneSAT and MethaneAIR activities was provided in part by Anonymous, Arnold Ventures, The Audacious Project, Ballmer Group, Bezos Earth Fund, The Children's Investment Fund Foundation, Heising-Simons Family Fund, King Philanthropies, Robertson Foundation, Skyline Foundation and Valhalla Foundation. For a more complete list of funders, please visit www.methanesat.org. Seed funding for development of the MethaneAIR sensor was provided by Harvard University, the Smithsonian Institution, and the Stonington Fund. We are grateful to several colleagues and teams for their efforts in some of the underlying data collection/processing including Jonathan Franklin, Jasna Pittman, MethaneSAT data processing, production operations teams.

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
