# Peer review of "Assessment of methane emissions from US onshore oil and gas production using MethaneAIR measurements"

_EGUsphere, 2025_

## Author Comment (AC1)

**Author response to reviewer comments**

**Anonymous Reviewer #1**

The authors quantify methane emissions from 12 US oil and gas basins using methane column observations from 32 MethaneAIR flights in 2023. These 12 basins accounted for 70% of total onshore oil and gas production in the contiguous United States in 2023. The authors estimate both total and sector-specific (oil + gas) emissions for each basin. They use a novel two-step regional flux inversion approach that first quantifies large point sources and then diffuse area emissions via Bayesian inverse analysis with the Stochastic Time-Inverted Lagrangian Transport (STILT) model. Emission contributions from non-oil and gas sources are estimated using sectoral emission estimates from a collection of previous top-down and bottom-up studies. The authors compare their regional estimates of methane emissions and loss rates with 16 previous studies and find generally good agreement.

We thank the reviewer for their helpful comments and suggestions. We have addressed the reviewer comments and incorporated them in the revised manuscript and hope the following responses address their concerns. Our point by point responses to all comments are below in blue, with the page numbers corresponding to the revised manuscript with tracked changes included.

The manuscript is well-written and a good fit for ACP. I recommend that it be accepted for publication with revisions to address the following comments and questions:

I think a more detailed description of the flux inversion methodology is needed, ideally in the main text. It is a novel approach and the most critical part of the analysis. Can more information be provided? For example, it would be helpful to know more about the modeling of the "boundary inflow", the numerical solution to the inverse problem, the approach to calculating column sensitivities with STILT (presumably based on MethaneAIR retrieval averaging kernels), and how/why GFS and HRRR meteorology are combined to drive STILT.

Response-We have substantially expanded Section S1 in the Supplement to describe the inversion framework in detail. We added methodological details and equations for the forward model, the background, the boundary inflow, and the solution to the inverse problem.

We clarified that independent Jacobians were computed using 1) GFS meteorology, and 2) HRRR meteorology where possible. These were used to diagnose potential issues with excessive transport error (diverging cases were excluded by QA/QC), and in the estimation of uncertainty. The following text is now included in Section S1:

Section S1, p. 2-4: *"With discrete sources ($s_{discrete}$) fixed, the inverse model fits a gridded field of dispersed area source emission rates ($s_{dispersed}$) to account for the balance of the methane enhancement. A gridded field of emission rates in the domain of interest, ($s_{reported} = s_{discrete} + s_{dispersed}$), and "pseudo-emission" rates in the upwind boundary inflow region ($s_{inflow}$) are fitted to observed column-averaged dry-air mole fractions of methane (XCH₄), z, linked by a Jacobian (H) plus a field of background concentrations (b) (Equation S1).*

$$z = H(s_{discrete} + s_{dispersed} + s_{inflow}) + b \qquad\qquad (Eq.\ S1)$$

The inversion enforces non-negative fluxes and exact conservation of the observed methane mass to maintain physical realism and applies Tikhonov regularization to promote spatial smoothness and mitigate the sensitivity of the hybrid framework to transport errors and measurement noise. We solve for the non-negative emission field ($s = s_{discrete} + s_{dispersed} + s_{inflow}$) that reproduces the MethaneAIR enhancements:

$$J_s = \|H_s - z - b\|^2 + \lambda^2 \|L(s)\|^2 \qquad\qquad (Eq.\ S2)$$
$$st. \quad s \geq 0, \quad w^T(H_s) = M,$$

where:
$L$ – first-order spatial difference operator enforcing smoothness
$\lambda$ – Tikhonov regularization strength
$M$ – total methane mass enhancement in the domain (kg $CH_4$)
$w$ – air-mass weights converting ppb to methane mass

$XCH_4$ observations were aggregated to 0.01° x 0.01° while preserving their location in time (allowing for overlapping observations from successive flight tracks). Aggregated grid cells at least 50% covered with data that passed all QA/QC flags were included in the analysis.

The Jacobian was computed using the Stochastic Time-Inverted Lagrangian Transport (STILT) model (Fasoli et al., 2018; Lin et al., 2003), which simulates the sensitivity of $XCH_4$ observations to sources on the ground by propagating air parcel trajectories backwards in time. The Jacobian was computed on a 0.01° x 0.01° grid over a 10° x 10° domain around the center of the flight with trajectories long enough to fully exit the domain or include the previous day's boundary layer (28 hours backtime). Where possible, the Jacobian was computed twice, 1) with STILT driven by meteorological data from the operational Global Forecast System (GFS) model and 2) with STILT driven by meteorology from the High-Resolution Rapid Refresh (HRRR) model. Meteorological data was provided by the NOAA ARL meteorological archives in ARL format (https://www.ready.noaa.gov/archives.php). STILT was run as a column receptor, with a receptor placed at every layer of the meteorological input from the surface to 3x the planetary boundary layer height (above which we assume the footprint is always 0). STILT footprints for every layer are integrated with weights representing the fraction of the total atmospheric column of dry air represented, with the mean averaging kernel for MethaneAIR.

The background concentration field represents the synoptic-scale, topographically varying component of the $XCH_4$ observations. We fit a field of background $XCH_4$ concentrations given by the MethaneAIR L2 prior (Chan Miller et al., 2024) from below, such that the reflected distribution of concentrations below background have a variance that matches the instrument precision. The MethaneAIR L2 prior forms a surface that varies realistically with topography in accordance with the vertical distribution of methane in the atmosphere from GEOS-FP Reanalysis (Rienecker et al., 2008) and the high-resolution digital elevation map tiles from Amazon Web Services (Larrick et al., 2020). Emissions are reported in a truncated domain of interest within the concave hull of the observations.

Boundary inflow "pseudo-emissions" are the component of the dispersed area source emissions inside the full 10° x 10° domain but outside the domain of interest. We refer to them as "pseudo-emissions" since they represent any source of sub-synoptic scale variation in the inflowing methane field, whether from mesoscale background variation or inflow of sources just outside the domain of interest.

*Discrete sources are fixed in the area source inversion, fixing emissions in a 0.01° x 0.01° area, which approximates the effective representative area of the DI. This places trust in the well-tested point-source specific algorithm to do the best job at quantifying point source emissions and uses the Jacobian to ensure the complete mass of methane from the point sources are accounted for. The alternative method of plume-masking is inconsistent between methodologies and inevitably undercounts the contribution of the point sources when they fall below detectable concentrations. The inclusion of fixed discrete sources in the inverse model makes this a constrained regularized optimization rather than Bayesian inference, and so we report only the optimal solution and do not assign probabilistic confidence intervals. It is important to note that this is non-Bayesian to prevent over-interpretation of a "posterior" estimate, which would be invalid due to data re-use.*

*The inverse problem is then solved numerically using projected, limited memory, bounded Broyden-Fletcher-Goldfarb-Shanno (L-BFGS-B) algorithm. The solution is initialized to a flat field that satisfies the mass constraint. Subsequent proposals are constrained to be non-negative and satisfy the mass constraint."*

What happens to the flux inversion if the modeled wind direction for a point source is wrong? In principle a flux dipole could arise, but it is mentioned in the SI that the non-negativity constraint in the inversion helps prevent that. How well does that work, and do any other biases manifest?

Response-When the modeled wind direction is wrong, plumes downwind of point sources are poorly modeled and a "plume shadow" or "dipole" effect is induced (a dipole is technically the result if non-negativity is not enforced). In cases where the modeled plume does not overlap the observed plume, the Jacobian is excluded by QA/QC. If both GFS and HRRR Jacobians are excluded by QA/QC, then the flight is excluded by QA/QC. There will always be some transport error, which is exacerbated where there are steep gradients in methane enhancements, most notably where there are distinct plumes. This is not only because of error in the mean wind direction, but also because of variations in XCH4 that cannot be resolved by the meteorological model (i.e., large eddies).

The inverse model uses a hard mass constraint Tikhonov regularization (more information about this has been added to the Supplement Section S1 – see above response for added text). This hard mass constraint ensures that the total excess methane in the observed atmosphere is modelled, and the spatial allocation of the inverse model distributes area emissions spatially throughout the region. The impact of wind error in the vicinity of large point sources is then to re-distribute other emissions sources around the plume, with minimal effect on the total, as the residence time of air is mostly unchanged by this perturbation.

How are point sources detected prior to the diffuse flux inversion? Is the process automated, semi-automated, or manual?

Response- Point sources are detected using an automated threshold-based method, with manual QA/QC prior to their inclusion in our analysis. The process is described in detail in Chulakadabba et al. 2023 and Warren et al. 2025 and validated in controlled release experiments (El Abbadi et al. 2024; Chulakadabba et al. 2023). We have expanded Section S1 in the Supplement as follows:

Section S1, p.2: *"For each MethaneAIR flight, discrete point source emissions (with methane emission rates > ~200 kg/hr), are detected using an automated threshold-based method with manual QA/QC prior to their inclusion in our analysis and subsequently quantified using a divergence integral (DI) method (Chulakadabba et al., 2023; Warren et al., 2025). The plume detection method first calculates the flux divergence for 600 m x 600 m squares tiled across*

*the scene, using High-Resolution Rapid Refresh (HRRR) wind fields and the divergence integral method (Chulakadabba et al., 2023) to calculate the flux through each square. In the gridded flux product, hotspots were identified with a thresholding method as potential plume origins. At each flux hotspot, we found $XCH_4$ clumps with a given number of contiguous pixels above a threshold value to create a mask of the plume. We calculated the major axis of the $XCH_4$ mask and took the upwind end of the major axis (using the HRRR wind direction) to be the plume origin (Warren et al., 2025). This system has been validated with controlled release experiments (Chulakadabba et al., 2023; El Abbadi et al., 2024), and is explained in greater detail in Warren et al., 2025."*

305-310: There is some redundant content in this passage.

Response- This passage importantly describes both the total methane emissions estimated by MethaneAIR as well as the oil and gas only methane emissions estimated by MethaneAIR, and how these two different estimates compare to the EPA totals. While the discrepancy between the overall total and oil and gas total is similar, we believe including both estimates and comparisons is valuable, as it shows that the oil and gas sector is likely the main contributor to underreported emissions in the GHGI.

314: 0.17 kg CH4/GJ from MethaneAIR is very similar to 0.18 kg CH4/GJ from IEA. Is it expected to be much lower? Perhaps this passage can be clarified.

Response- We modified the text to add clarity to the comparison between the MethaneAIR and IEA intensity estimates, as there are other important differences between the two in addition to the use of gross vs. marketed gas production that should be mentioned. Considering these other factors, we do not expect the IEA value to necessarily be lower than our MethaneAIR estimate, so we modified the text as follows:

L312-315: *"The estimated energy-normalized methane intensity of 0.17 kg $CH_4$/GJ is comparable to the upstream methane intensity of 0.18 kg $CH_4$/GJ for the entire US reported by the IEA for 2024 (IEA, 2025), however it should be noted that their estimate is calculated using marketed oil and gas production, whereas our estimate uses gross production and includes methane emissions from the entire oil and gas sector (i.e., not just upstream)."*

319: I believe Figure 6 is mislabeled here—the passage seems to refer to Figure 5.

Response- We double checked the caption for Figure 6 and confirmed that there is no labelling error, it describes the comparison of MethaneAIR derived loss rates to other measurement-based loss rates from previous literature. Note that there are some similarities in the features for Figures 5 and 6 (e.g., the grey shaded area and dashed lines), hence the similar descriptions in the captions.

Figures 5 and 6:
In which cases are the MethaneAIR and previous estimates for the same domain? Is the Zhang et al. result for the Permian spatially resampled to the flight domain? Those authors reported a loss rate of 3.7%, much higher than the <2% shown in Figure 6, so I assume so. It would be helpful to mark on the plots whether or not the previous results reflect spatial resampling.

Response- We added an asterisk to the x-axis labels and expanded the caption text to note which previous estimates correspond to the exact domain, and which ones correspond to similar/overlapping areas. Regarding the Zhang et al., 2020 reported loss rate, the difference is related to the domains - their

reported loss rate reflects their entire study domain (i.e., the whole Permian basin) whereas the loss rate we show in the figure was computed based on Zhang et al.'s reported emissions and production volumes within the MethaneAIR flight domain which is a subset of the Permian basin.

Why are the x labels in the figures not identical? There are fewer bars in some subplots of Figure 6 than Figure 5.

Response- There are a different number of bars in the figures because some studies only reported total methane emissions and did not report methane loss rates or the necessary information (e.g., gas production volumes at the time of measurement) for us to compute it for our analysis. In these cases, they are included in Figure 5 but not included in Figure 6.

Why do the inter-study differences in methane emissions not more closely match the differences in methane loss rate? Two examples of this: the Peischl bars in the Barnett subplots show much higher emissions than MethaneAIR but very similar loss rate, and the MethaneAIR bars in the Permian subplots show better agreement with previous studies for emissions than loss rate.

Response- Absolute emissions can have more variability due to changes in activity/production levels over time, whereas methane loss rates are often more stable over time as they normalize emissions by production. This is likely contributing to the observed difference in the Peischl et al., 2018 study in the Barnett, as those measurements were collected in 2015 when oil and gas production was much higher than when the MethaneAIR flights took place in 2023. For the Permian, the time of measurement for many of the studies is similar, likely contributing to the better agreement in total emissions. We also argue that the loss rates show similarly good agreement for the Permian (<1% difference across all studies).

**References:**

[revised manuscript text omitted]

---

## Author Comment (AC2)

**Author response to reviewer comments**

**Anonymous Reviewer #2**

The authors present results from many MethaneAIR flights performed in the United States, primarily to quantify oil&gas emissions across major basins. Overall this represents a tremendous body of work to execute the flights, process data, and analyze for fluxes and the authors should be commended for this effort. The manuscript succinctly summarizes the results and shows general consistency in emission rates derived from previous observation studies (mass-balance, and satellite remote sensing). The authors employ what appears to be a novel way to calculate methane fluxes from remote sensing observations, but the details are extremely light.

Response- We thank the reviewer for their helpful comments and edits, and hope the following responses address their concerns. Our point by point responses to all comments are below in blue, with the page numbers corresponding to the attached revised manuscript with tracked changes included.

Before I can accept for publication, considerable more detail needs to be included and justified. My comments are as follows:

1. Table 1. Is Basin Area the total area of the basin or total area flown? If total area, can you express in the same table how much of that area you flew with MethaneAIR?

Response- The basin area in Table 1 is indeed the total area, not the total area flown. We have removed this column in Table 1 as it is not used in our analysis. In Table 3 in the main text, we added the area (in $km^2$) covered by the MethaneAIR flights in each oil and gas basin. We also report the fraction of the basin's total production covered during the MethaneAIR flights, which is a more informative metric than the percent of the basin's area flown, since some basins have significant geographic coverage but production is concentrated in smaller regions which is where we would expect the majority of methane emissions to originate from.

Section S1. STILT.

2. How do you simulate columns with STILT? How many layers? Interpolate between layers? Use an averaging kernel? What is the averaging kernel?

Response- We have expanded the Supplement (Section S1) to add details about the column integration of the STILT model. In short, we ran STILT with 300 particles at the center of every meteorological model layer for each of the GFS and HRRR model up to 3x the planetary boundary layer height with integration using the fraction of total atmosphere column dry air and the mean MethaneAIR averaging kernel (see Chan Miller at al., 2024) as the weighting function. Section S1 now includes the following details:

Section S1, p.3: "*The Jacobian was computed using the Stochastic Time-Inverted Lagrangian Transport (STILT) model (Fasoli et al., 2018; Lin et al., 2003), which simulates the sensitivity of $XCH_4$ observations to sources on the ground by propagating air parcel trajectories backwards in time. The Jacobian was computed on a 0.01° x 0.01°*

*grid over a 10° x 10° domain around the center of the flight with trajectories long enough to fully exit the domain or include the previous day's boundary layer (28 hours backtime). Where possible, the Jacobian was computed twice, 1) with STILT driven by meteorological data from the operational Global Forecast System (GFS) model and 2) with STILT driven by meteorology from the High-Resolution Rapid Refresh (HRRR) model. Meteorological data was provided by the NOAA ARL meteorological archives in ARL format (https://www.ready.noaa.gov/archives.php). STILT was run as a column receptor, with a receptor placed at every layer of the meteorological input from the surface to 3x the planetary boundary layer height (above which we assume the footprint is always 0). STILT footprints for every layer are integrated with weights representing the fraction of the total atmospheric column of dry air represented, with the mean averaging kernel for MethaneAIR."*

Section S1. The calculation of the background is unclear.

Response- We have substantially expanded the Supplement (Section S1) to add details about the background concentration calculation. In short, the background model is a field of methane concentrations taken from the MethaneAIR L2 product (see Chan Miller et al., 2024), fitted from below to the observations, with an allowance for the instrument precision.

Section S1, p.3: "*The background concentration field represents the synoptic-scale, topographically varying component of the $XCH_4$ observations. We fit a field of background $XCH_4$ concentrations given by the MethaneAIR L2 prior (Chan Miller et al., 2024) from below, such that the reflected distribution of concentrations below background have a variance that matches the instrument precision. The MethaneAIR L2 prior forms a surface that varies realistically with topography in accordance with the vertical distribution of methane in the atmosphere from GEOS-FP Reanalysis (Rienecker et al., 2008) and the high-resolution digital elevation map tiles from Amazon Web Services (Larrick et al., 2020). Emissions are reported in a truncated domain of interest within the concave hull of the observations."*

3. Can you restate in terms of an equation, figure, or additional clarifying language? - "The background concentrations are given by a model..." - what model? STILT?

Response- We have expanded the Supplement (Section S1) to add details about the background concentration calculation. See above response for revised text added to Section S1.

4. "The boundary inflow is modeled using the Jacobian and emission rates outside the domain of observed concentrations." Where do you get these emissions? An inventory? Proper background quantification is so vital to robust inversions, this section needs to be much clearer.

Response- The boundary inflow is computed as pseudo-emissions outside the domain of interest but inside the total domain of the inversion. These are "pseudo-emissions" that represent actual emissions just outside the domain or mesoscale variability in the inflowing methane concentration. This section has been made clearer:

Section S1, p.3: "*Boundary inflow "pseudo-emissions" are the component of the dispersed area source emissions inside the full 10° x 10° domain but outside the domain of interest. We refer to them as "pseudo-emissions" since they represent any source of sub-synoptic scale variation in the inflowing methane field, whether from mesoscale background variation or inflow of sources just outside the domain of interest."*

Section S1. Point Sources.

5. Is the divergence integral method to calculate point sources applied at the 0.01 binning or at the native resolution? If you are binning, then you are certainly subtracting out more than point sources, as you are aggregating all true emission sources within that ~1km domain.

Response- We are indeed binning. The divergence integral integrates emissions from an effective area of approximately 1 km$^2$ for MethaneAIR. The divergence integral integrates all emissions in this area, and so fixing the emissions in this gridcell is the appropriate choice to best represent the computed discrete sources.

6. If you are not binning, how do you assess that model transport error correctly subtracts the influence of point sources? Do you have a quality control approach that ensures this? If the transport is wrong, then you risk not subtracting the point source component in your concentration field, which I can imagine will throw the inversion haywire and produce flux artifacts.

Response- See above response regarding binning. If the transport is substantially wrong such that the total emissions in a flight are substantially affected, the flight fails quantity control and is excluded from the analysis.

7. How do you determine an origin of the point source via the divergence integral method? It seems quite critical that you get the origin correct if you are forward simulating a concentration field with STILT.

Response- The plume detection method first calculates the flux divergence for 600 m x 600 m squares tiled across the scene, using HRRR wind fields and the divergence integral method to calculate the flux through each square. The upwind end of each plume shows a higher flux divergence than the downwind end; hotspots in the gridded flux product were identified with a thresholding method as potential plume origins. At each flux hotspot, we found XCH$_4$ clumps with a given number of contiguous pixels above a threshold value to create a mask of the plume. We calculated the major axis of the XCH$_4$ mask and took the upwind end of the major axis (using the HRRR wind direction) to be the plume origin. All plumes found using this method were reviewed manually by examining plume morphology, ground infrastructure, and correlation with albedo, and false positives were discarded.

We have modified the text in the Supplement (Section S1) to include more detail on the method, as follows:

Section S1, p.2: *"For each MethaneAIR flight, discrete point source emissions (with methane emission rates > ~200 kg/hr), are detected using an automated threshold-based method with manual QA/QC prior to their inclusion in our analysis and subsequently quantified using a divergence integral (DI) method (Chulakadabba et al., 2023; Warren et al., 2025). The plume detection method first calculates the flux divergence for 600 m x 600 m squares tiled across the scene, using High-Resolution Rapid Refresh (HRRR) wind fields and the divergence integral method (Chulakadabba et al., 2023) to calculate the flux through each square. In the gridded flux product, hotspots were identified with a thresholding method as potential plume origins. At each flux hotspot, we found XCH$_4$ clumps with a given number of contiguous pixels above a threshold value to create a mask of the plume. We calculated the major axis of the XCH$_4$ mask and took the upwind end of the major axis (using the HRRR wind direction) to be the plume origin (Warren et al., 2025). This system has been validated with controlled release experiments (Chulakadabba et*

*al., 2023; El Abbadi et al., 2024), and is explained in greater detail in Chulakadabba et al., 2023 and Warren et al., 2025."*

8. It's not obvious to the reader that subtracting a forward model simulated concentration field produces a preferred result, especially assuming some level of spatial aggregation (e.g., were you to run these inversions at 10-20km, scales that others perform satellite inversions, would this still be required?).

Response- Rather than subtracting a forward model simulated concentration field we fix the emissions in the gridcell where the discrete source was detected. This is the most accurate interpretation of the quantity measured by the divergence integral algorithm.

Section S1. Inverse Problem

9. What is the formulation of the inversion problem? An SI is a good place to put these equations to paper. The way it currently reads, the mass-balance constraint would just be the inverse of the Jacobian - i.e., s = (H^-1)y and the non-negativity constraint would be some sort of gradient descent (or something else?) algorithm that stops at zero. Not obvious from what's written. Are there other parameters that keep the solution from an overfit? They say there is no need for a prior, so not Bayesian I guess?

Response- We have substantially increased the amount of detail (revised text below) about the inverse model in the Supplement (Section S1).

Section S1, p. 2-4: *"With discrete sources ($s_{discrete}$) fixed, the inverse model fits a gridded field of dispersed area source emission rates ($s_{dispersed}$) to account for the balance of the methane enhancement. A gridded field of emission rates in the domain of interest, ($s_{reported} = s_{discrete} + s_{dispersed}$), and "pseudo-emission" rates in the upwind boundary inflow region ($s_{inflow}$) are fitted to observed column-averaged dry-air mole fractions of methane (XCH₄), z, linked by a Jacobian ($H$) plus a field of background concentrations ($b$) (Equation S1).*

$$z = H(s_{discrete} + s_{dispersed} + s_{inflow}) + b \qquad \text{(Eq. S1)}$$

*The inversion enforces non-negative fluxes and exact conservation of the observed methane mass to maintain physical realism and applies Tikhonov regularization to promote spatial smoothness and mitigate the sensitivity of the hybrid framework to transport errors and measurement noise. We solve for the non-negative emission field ($s = s_{discrete} + s_{dispersed} + s_{inflow}$) that reproduces the MethaneAIR enhancements:*

$$J_s = \|H_s - z - b\|^2 + \lambda^2 \|L(s)\|^2 \qquad \text{(Eq. S2)}$$
$$st. \quad s \geq 0, \quad w^T(H_s) = M,$$

*where:*
*L – first-order spatial difference operator enforcing smoothness*
*λ – Tikhonov regularization strength*
*M – total methane mass enhancement in the domain (kg CH₄)*
*w – air-mass weights converting ppb to methane mass*

*XCH₄ observations were aggregated to 0.01° x 0.01° while preserving their location in time (allowing for overlapping observations from successive flight tracks). Aggregated grid cells at least 50% covered with data that passed all QA/QC flags were included in the analysis.*

*The Jacobian was computed using the Stochastic Time-Inverted Lagrangian Transport (STILT) model (Fasoli et al., 2018; Lin et al., 2003), which simulates the sensitivity of XCH₄ observations to sources on the ground by propagating air parcel trajectories backwards in time. The Jacobian was computed on a 0.01° x 0.01° grid over a 10° x 10° domain around the center of the flight with trajectories long enough to fully exit the domain or include the previous day's boundary layer (28 hours backtime). Where possible, the Jacobian was computed twice, 1) with STILT driven by meteorological data from the operational Global Forecast System (GFS) model and 2) with STILT driven by meteorology from the High-Resolution Rapid Refresh (HRRR) model. Meteorological data was provided by the NOAA ARL meteorological archives in ARL format (https://www.ready.noaa.gov/archives.php). STILT was run as a column receptor, with a receptor placed at every layer of the meteorological input from the surface to 3x the planetary boundary layer height (above which we assume the footprint is always 0). STILT footprints for every layer are integrated with weights representing the fraction of the total atmospheric column of dry air represented, with the mean averaging kernel for MethaneAIR.*

*The background concentration field represents the synoptic-scale, topographically varying component of the XCH₄ observations. We fit a field of background XCH₄ concentrations given by the MethaneAIR L2 prior (Chan Miller et al., 2024) from below, such that the reflected distribution of concentrations below background have a variance that matches the instrument precision. The MethaneAIR L2 prior forms a surface that varies realistically with topography in accordance with the vertical distribution of methane in the atmosphere from GEOS-FP Reanalysis (Rienecker et al., 2008) and the high-resolution digital elevation map tiles from Amazon Web Services (Larrick et al., 2020). Emissions are reported in a truncated domain of interest within the concave hull of the observations.*

*Boundary inflow "pseudo-emissions" are the component of the dispersed area source emissions inside the full 10° x 10° domain but outside the domain of interest. We refer to them as "pseudo-emissions" since they represent any source of sub-synoptic scale variation in the inflowing methane field, whether from mesoscale background variation or inflow of sources just outside the domain of interest.*

*Discrete sources are fixed in the area source inversion, fixing emissions in a 0.01° x 0.01° area, which approximates the effective representative area of the DI. This places trust in the well-tested point-source specific algorithm to do the best job at quantifying point source emissions and uses the Jacobian to ensure the complete mass of methane from the point sources are accounted for. The alternative method of plume-masking is inconsistent between methodologies and inevitably undercounts the contribution of the point sources when they fall below detectable concentrations. The inclusion of fixed discrete sources in the inverse model makes this a constrained regularized optimization rather than Bayesian inference, and so we report only the optimal solution and do not assign probabilistic confidence intervals. It is important to note that this is non-Bayesian to prevent over-interpretation of a "posterior" estimate, which would be invalid due to data re-use.*

*The inverse problem is then solved numerically using projected, limited memory, bounded Broyden-Fletcher-Goldfarb-Shanno (L-BFGS-B) algorithm. The solution is initialized to a flat field that satisfies the mass constraint. Subsequent proposals are constrained to be non-negative and satisfy the mass constraint."*

10. The authors do not provide any sort of metric of goodness of fit (e.g., H * s_hat plotted against y) or information content from the retrieval. It's fairly common practice to show how well your model around the optimal emission state compares to observations. It's also common to show information content

metrics (e.g., degrees of freedom for signal, model-resolution matrices, etc) for inversions, but given there's not an explicit inverse formulation in paper, it's hard to know if that would be feasible.

Response- We have added significant details about the inversion methodology in the Supplement Section S1. As included in our further details in Section S1 and in our responses, this analysis is based on a constrained regularized optimization rather than Bayesian inference. We have also added a plot (Figure S5) showing the observed vs modeled $XCH_4$ enhancement for MethaneAIR flights.

11. Table S5. Please provide citations for literature based estimates, perhaps as an additional column in the table or a footnote.

We added the citations for the literature-based estimates in Table S5.

**References:**

Chan Miller, C., Roche, S., Wilzewski, J. S., Liu, X., Chance, K., Souri, A. H., Conway, E., Luo, B., Samra, J., Hawthorne, J., Sun, K., Staebell, C., Chulakadabba, A., Sargent, M., Benmergui, J. S., Franklin, J. E., Daube, B. C., Li, Y., Laughner, J. L., Baier, B. C., Gautam, R., Omara, M., and Wofsy, S. C.: Methane retrieval from MethaneAIR using the $CO_2$ proxy approach: a demonstration for the upcoming MethaneSAT mission, Atmospheric Measurement Techniques, 17, 5429–5454, https://doi.org/10.5194/amt-17-5429-2024, 2024.

Chulakadabba, A., Sargent, M., Lauvaux, T., Benmergui, J. S., Franklin, J. E., Chan Miller, C., Wilzewski, J. S., Roche, S., Conway, E., Souri, A. H., Sun, K., Luo, B., Hawthrone, J., Samra, J., Daube, B. C., Liu, X., Chance, K., Li, Y., Gautam, R., Omara, M., Rutherford, J. S., Sherwin, E. D., Brandt, A., and Wofsy, S. C.: Methane point source quantification using MethaneAIR: a new airborne imaging spectrometer, Atmospheric Measurement Techniques, 16, 5771–5785, https://doi.org/10.5194/amt-16-5771-2023, 2023.

El Abbadi, S. H., Chen, Z., Burdeau, P. M., Rutherford, J. S., Chen, Y., Zhang, Z., Sherwin, E. D., and Brandt, A. R.: Technological Maturity of Aircraft-Based Methane Sensing for Greenhouse Gas Mitigation, Environ. Sci. Technol., 58, 9591–9600, https://doi.org/10.1021/acs.est.4c02439, 2024.

Fasoli, B., Lin, J. C., Bowling, D. R., Mitchell, L., and Mendoza, D.: Simulating atmospheric tracer concentrations for spatially distributed receptors: updates to the Stochastic Time-Inverted Lagrangian Transport model's R interface (STILT-R version 2), Geoscientific Model Development, 11, 2813–2824, https://doi.org/10.5194/gmd-11-2813-2018, 2018.

Larrick, G., Tian, Y., Rogers, U., Acosta, H., and Shen, F.: Interactive Visualization of 3D Terrain Data Stored in the Cloud, in: 2020 11th IEEE Annual Ubiquitous Computing, Electronics & Mobile Communication Conference (UEMCON), 2020 11th IEEE Annual Ubiquitous Computing, Electronics & Mobile Communication Conference (UEMCON), 0063–0070, https://doi.org/10.1109/UEMCON51285.2020.9298063, 2020.

Lin, J. C., Gerbig, C., Wofsy, S. C., Andrews, A. E., Daube, B. C., Davis, K. J., and Grainger, C. A.: A near-field tool for simulating the upstream influence of atmospheric observations: The Stochastic Time-Inverted Lagrangian Transport (STILT) model, Journal of Geophysical Research: Atmospheres, 108, https://doi.org/10.1029/2002JD003161, 2003.

Rienecker, M. M., Suarez, M. J., Todling, R., Bacmeister, J., Takacs, L., Liu, H.-C., Gu, W., Sienkiewicz, M., Koster, R. D., Gelaro, R., Stajner, I., and Nielsen, J. E.: The GEOS-5 Data Assimilation System-Documentation of Versions 5.0.1, 5.1.0, and 5.2.0, 2008.

Warren, J. D., Sargent, M., Williams, J. P., Omara, M., Miller, C. C., Roche, S., MacKay, K., Manninen, E., Chulakadabba, A., Himmelberger, A., Benmergui, J., Zhang, Z., Guanter, L., Wofsy, S., and Gautam, R.: Sectoral contributions of high-emitting methane point sources from major US onshore oil and gas producing basins using airborne measurements from MethaneAIR, Atmospheric Chemistry and Physics, 25, 10661–10675, https://doi.org/10.5194/acp-25-10661-2025, 2025.

---

## Author Response (AR2)

**Author response to reviewer comments**

**Anonymous Reviewer #2**

The authors have now included much more detail about their technical methodology to estimate emissions from dispersed sources. I have some specific questions about this new section, but otherwise the manuscript is much improved.

We thank the reviewer for their helpful comments on the revised manuscript. We addressed the reviewer comments and revised the manuscript in the SI section where the methodology is described. Our point by point responses to all comments are below in blue.

1. Section S1. Equations in SI - make sure you are using consistent notation for vectors vs matrices. H is a matrix in Eq S1 and s is a vector, correct? Please use consistent notation

Thank you for identifying this error. We have corrected the equations in the SI (section S1) to use consistent notation.

2. Section S1. "The inclusion of fixed discrete sources in the inverse model makes this a constrained regularized optimization rather than Bayesian inference, and so we report only the optimal solution and do not assign probabilistic confidence intervals. It is important to note that this is non-Bayesian to prevent overinterpretation of a "posterior" estimate, which would be invalid due to data re-use." I'm not sure I agree with the first sentence as written. What makes this non-Bayesian is the fact that you are applying regularization parameters and hard constraints in your optimization protocol, not because you have discrete point sources as a state element in the equation. For the second sentence, as written this is confusing. If you have truly separated discrete elements from dispersed elements, and fixed the discrete variable, I don't understand how data would be "re-used" if you were to apply a bayesian method, at least in a way different than a Tikhonov formulation. Please clarify.

We thank the reviewer for this clarification and agree that the original wording was imprecise. We have clarified the text accordingly in section S1.

First, we agree that the inclusion of discrete point sources as state elements does not, by itself, render an inverse problem non-Bayesian. The non-Bayesian nature of our approach arises from the way the diffuse inversion is formulated and solved: namely, as a constrained, regularized optimization with a fixed regularization parameter and hard constraints (non-negativity and a mass constraint), rather than as a probabilistic inference with explicitly defined priors, likelihoods, and uncertainty propagation.

Second, we appreciate the opportunity to clarify our use of the term "data re-use." In our framework, discrete source emissions are first estimated using a divergence-integral (DI) method applied to a subset of the MethaneAIR $XCH_4$ observations. These discrete emission rates are then fixed and treated as known quantities in the subsequent optimization for dispersed emissions. The dispersed emissions are inferred using an objective function that is evaluated against the full $XCH_4$ observation set, which may include observations that were previously used to estimate the discrete sources.

As a result, the same observations can influence both the discrete-source estimates and the dispersed-source solution, without explicit propagation of uncertainty from the discrete-source step into the dispersed inversion. A fully Bayesian treatment would require either (i) joint inference of discrete and dispersed emissions within a single probabilistic framework, or (ii) propagation of uncertainty from the discrete-source estimates into the dispersed inversion. Because neither is done here, interpreting the dispersed solution as a Bayesian posterior would be incomplete. For this reason, we deliberately adopt a non-Bayesian framing and report only the deterministic optimal solution of the constrained optimization, rather than probabilistic confidence intervals or posterior distributions. We have revised section S1 to reflect this clarification and to remove the implication that discrete sources themselves make the problem non-Bayesian, as follows:

Section S1, p. 3: *"The same observations can influence both the discrete-source estimates and the dispersed-source solution, without explicit propagation of uncertainty from the discrete-source step into the dispersed inversion. A fully Bayesian treatment would require either (i) joint inference of discrete and dispersed emissions within a single probabilistic framework, or (ii) propagation of uncertainty from the discrete-source estimates into the dispersed inversion. Because neither is done here, interpreting the dispersed solution as a Bayesian posterior would be incomplete. Therefore, we deliberately adopt a non-Bayesian framing and report only the deterministic optimal solution of the constrained optimization, rather than probabilistic confidence intervals or posterior distributions."*

3. Section S1. I have a question regarding lingering concentrations from discrete elements: "Discrete sources are fixed in the area source inversion, fixing emissions in a 0.01° x 0.01° area, which approximates the effective representative area of the DI." As I understand, the DI approach assumes that for a discrete source, there is downwind concentration that hits the boundary of a square tile. The implication is that there is likely additional concentration that extends beyond the 600 m tile, and potentially beyond a 0.01 degree grid cell that encloses that tile if the plume is sufficiently large. By not segmenting, accounting, and

attributing those concentration enhancements to the discrete sources from which they originate, those enhancements now get lumped/used in the inversion for s_dispersed. Do you account for that potential source of bias and how? Or clarify how this does not incur potential bias in the result for s_dispersed.

We thank the reviewer for bringing to light this important clarification. In our framework, discrete sources are incorporated as fixed state elements in the inverse problem by fixing the emission rate of the grid cell containing the detected source. Consequently, downwind concentration residuals associated with discrete sources are, in principle, available to be fit by the diffuse emission field. This design choice was intentional and reflects a prioritization of accuracy of total emissions over the spatial distribution of the dispersed emission field. Our paper is entirely focused on total emissions from the individual regions.

If we attempted to segment the observable downwind enhancements due to the discrete sources, we would inevitably 1) underestimate the total enhancement from the discrete source where those enhancements fall below detectability (i.e., under-estimate the extent of the plume due to observation noise), and/or 2) overestimate the total enhancement by lumping other sources into the segmented downwind enhancement. By fixing the discrete source to the DI estimate, we allow the Jacobian and hard mass constraint to close the total methane emissions budget.

The diffuse emission inversion is formulated to explain the observed enhancement field subject to non-negativity, smoothness regularization, and a hard mass constraint. Under these constraints, any residual signal downwind of a fixed discrete source is preferentially distributed smoothly over the surrounding area rather than producing localized artifacts. As a result, plumes from discrete sources are effectively absorbed into the local diffuse field with only small downwind effects and without introducing spurious secondary discrete sources.

We acknowledge that this approach may attribute some fraction of discrete-source plume tails to the diffuse component, and we have clarified this explicitly in the revised SI. However, this attribution does not affect the basin-integrated emission total, which is constrained by the hard mass constraint.

Section S1, p. 3: *"We acknowledge that this approach may attribute some fraction of discrete-source plume tails to the diffuse component, but this does not affect the basin-integrated emission total, which is constrained by the hard mass constraint."*

4. Section S1. "The solution is initialized to a flat field that satisfies the mass constraint. Subsequent proposals are constrained to be non-negative and satisfy the mass constraint. The two constraints regularize the solution to prevent overfitting." Reducing overfitting in a Tikhonov scheme is also accomplished by the regularization parameter lambda. I don't see anything in the SI of how you choose that parameter, or is that parameter chosen as a result of the other constraints? Please clarify especially with respect to these other constraints.

The constraints and the Tikhonov term serve complementary roles: non-negativity enforces physical feasibility; the hard mass constraint enforces domain-scale consistency; and $\lambda$ controls spatial roughness and suppresses fitting of retrieval/transport noise at small scales. A value of $\lambda=0.5$ was selected using an L-curve criterion computed over a set of representative scenes. We have added this information in the SI:

Section S1, p. 3: *"The non-negativity and mass-balance constraint are complementary to the Tikhonov term such that non-negativity enforces physical feasibility, the hard mass constraint enforces domain-scale consistency, and $\lambda$ controls spatial roughness and suppresses fitting of retrieval/transport noise at small scales. A value of $\lambda=0.5$ was selected using an L-curve criterion computed over a set of representative scenes."*